# Medical Management versus PACK-CXL in Dogs with Infectious Keratitis: A Randomized Controlled Trial Protocol

**DOI:** 10.3390/ani12202862

**Published:** 2022-10-20

**Authors:** Malwina E. Kowalska, Farhad Hafezi, Simon A. Pot, Sonja Hartnack

**Affiliations:** 1Ophthalmology Section, Equine Department, Vetsuisse Faculty, University of Zurich, CH-8057 Zurich, Switzerland; 2Section of Epidemiology, Vetsuisse Faculty, University of Zurich, CH-8057 Zurich, Switzerland; 3ELZA Institute, CH-8953 Dietikon, Switzerland

**Keywords:** corneal ulcer, block sequential design, canine

## Abstract

**Simple Summary:**

Infected corneal ulcers are a common and painful disease, usually caused by bacteria in dogs. Short-nosed dog breeds are predisposed to corneal ulcers. Treatment aims to eliminate bacteria, stop the enzymatic tissue melting process in the cornea, and allow for normal healing to resume. Success of treatment largely depends on the effectivity of the applied medications, possibly including antibiotics. In the context of globally growing antibiotic resistance, non-antibiotic-based treatment alternatives are desirable. Photoactivated Chromophore for Keratitis-Corneal Cross-linking (PACK-CXL) reduces enzymatic tissue melting and damages multiple targets within microorganisms, resulting in corneal stabilization and non-antibiotic-based elimination of bacteria. A confirmatory clinical study providing unbiased evidence of PACK-CXL effectiveness in dogs is currently lacking. We therefore aim to conduct such a study and determine whether PACK-CXL is a viable alternative to conventional state-of-the-art medical therapy for corneal ulcer treatment in dogs. Here we present the randomized controlled clinical trial study protocol. Registering trials favors sound statistical analysis and prevents publication bias. The trial results will be published after successful trial conclusion.

**Abstract:**

Infectious keratitis is a common and painful disease, usually caused by bacteria in dogs. Brachycephalic breeds are at increased risk. Despite medical therapy, enzymatic corneal melting can lead to ulcer perforation and globe loss. Treatment alternatives are needed due to an increase in antibiotic resistance and growing popularity of brachycephalic dogs. Photoactivated Chromophore for Keratitis-Corneal Cross-linking (PACK-CXL) reduces enzymatic collagenolysis and damages multiple targets within microorganisms, resulting in corneal tissue stabilization and elimination of bacteria, irrespective of their antibiotic resistance status. A randomized controlled trial providing evidence of PACK-CXL effectiveness in dogs is lacking. We aim to determine whether PACK-CXL is a viable alternative to conventional medical therapy for canine infectious keratitis. Two hundred-and-seventy client-owned dogs with presumed infectious keratitis will be allocated to two equally sized treatment groups (PACK-CXL or medical therapy) in a masked, randomized, controlled, multicenter trial in eleven clinics. The primary outcome measure is treatment success defined as complete epithelial closure within 28 days. The sample size is based on a group sequential design with two interim analyses, which will be overseen by a Data Safety and Monitoring Board. Ethical approvals have been obtained. The study protocol is preregistered at preclinicaltrials.eu. Publishing trial protocols improves study reproducibility and reduces publication bias.

## 1. Introduction

A clinical trial protocol provides the basis for study planning, execution, and reporting. It is critical to address scientific, ethical, and safety issues before a trial begins. Moreover, publicly available protocols increase awareness of ongoing trials, which may improve enrollment. Furthermore, trial pre-registration and/or a registered report reduces reporting and publication bias, thus improving replicability. Our protocol is reported according to the Standard Protocol Items: Recommendations for Interventional Trials (SPIRIT), an evidence-based guideline for the minimum content of a clinical trial protocol [1], and CONSORT guidelines [2].

Infectious keratitis is a painful, and potentially blinding disease in all species, with a prevalence of 0.8% in dogs [3,4,5]. Bacteria are the most commonly involved microorganism in dogs [3]. During the course of the disease, the inflammatory response to the infection activates proteolytic collagen-dissolving enzymes in the corneal stroma, resulting in corneal stromal tissue loss, corneal ulcer development and deepening, and possibly corneal perforation and loss of vision [3,4,6,7]. The rapid initiation of a treatment that inhibits corneal tissue degradation and eliminates pathogens early in the disease is essential for treatment success. The current gold standard non-surgical infectious keratitis treatment in dogs is intensive medical therapy, involving the initially frequent application of anti-collagenase and antibiotic eye drops every 1 to 2 h [4,8]. Treatment success rates vary from 70 to 75% (based on clinical unpublished data from the University of Zurich, Zurich, Switzerland, and the Animal Health Trust, Newmarket, UK). Nevertheless, there is a global need to develop alternative treatment methods due to: Increasing numbers of dogs suffering from infectious keratitis as a result of the increasing popularity of brachycephalic breeds which are predisposed to develop corneal ulcers, with an odds ratio (OR) of 6 to 11 compared to typical meso- or dolichocephalic dogs [5,9];Growing concerns regarding increasing antibiotic resistance amongst ocular pathogens [10,11,12,13,14,15,16,17,18];Compliance problems as a result of treatment frequency.

Photoactivated Chromophore for Keratitis-Corneal Cross-linking (PACK-CXL) is a photodynamic therapy that presents an alternative to standard medical infectious keratitis therapy.

During the PACK-CXL procedure, a chromophore solution (e.g., riboflavin) is first applied to the corneal surface until tissue saturation. Second, the cornea is exposed to irradiation (e.g., 365 nm wavelength UV-A). This induces the photochemical formation of cross-links within and on the surface of collagen fibers and between collagen fibers and proteoglycans in the extracellular matrix [19]. PACK-CXL thus improves corneal resistance to enzymatic digestion, arresting corneal stromal tissue degradation [20,21]. Importantly, at the same time, PACK-CXL shows antimicrobial properties by damaging multiple targets within microorganisms. The activated chromophore interacts with microbial nucleic acids, inhibiting replication [22], and released reactive oxygen free radicals damage the microbial cell wall [23,24,25].

An important advantage of PACK-CXL is that antibiotic-resistant and non-resistant bacteria are equally sensitive [24] and that pathogens are unable to develop resistance to PACK-CXL compared to traditional antibiotic therapy [26,27,28]. If broadly used as treatment for infectious keratitis, PACK-CXL may help to reduce bacterial resistance to topical antibiotics. This aspect is equally important for canine and human ocular health.

The first clinical studies that described the use and beneficial effects of PACK-CXL in companion animals, including non-randomized clinical trials in cats and dogs, were published in 2014 [29,30,31]. Systematic reviews and meta-analyses of evidence derived from case reports and small trials involving human patients are available [32,33,34]. A multicenter randomized controlled trial in human patients with early stage infectious keratitis has been published. A single standalone PACK-CXL treatment resulted in comparable healing times and treatment success rates when compared to medical treatment over several weeks [35]. Canine cases are typically presented with advanced disease, which makes the extrapolation of the results from many trials involving human patients problematic, emphasizing the need for a randomized controlled trial in dogs.

Unbiased evidence of PACK-CXL superiority to standard therapy is essential to encourage medical professionals to use it as a replacement of or an adjunctive to routine medical treatment. When evidence is available, general veterinary practitioners are likely to adopt PACK-CXL, since it is simple, relatively inexpensive, and works against antibiotic-resistant and non-resistant bacteria. Unlike standard care, PACK-CXL can be administered in a single application, and patients may require far less frequent medications and re-examinations. The availability of PACK-CXL in general practice will rapidly increase early access for a wider public to effective infectious keratitis treatment. This will further increase treatment success rates, as cases treated at an early disease stage seem to have a better overall prognosis [32,36].

Treatment effectiveness may not be the only information owners want when deciding on the treatment option. Since pet owners are not required to purchase pet health insurance, the veterinary service and medication costs (initial visit, procedure, recheck visits, medications etc.) may be a crucial consideration. The PACK-CXL procedure could require either sedation or general anesthesia, which can initially result in high expenditures. However, fewer visits and medications may result in lower treatment costs. It is uncertain which treatment for infectious keratitis is less expensive.

The proposed randomized, controlled, multicenter trial with two equally sized, parallel treatment groups will answer the question whether treatment of presumed infectious keratitis in dogs with ultraviolet-A corneal cross-linking with riboflavin as photosensitizer will result in higher treatment success rates compared to current state-of-the-art medical therapy. A sufficiently powered clinical trial providing unbiased evidence of PACK-CXL effectiveness in dogs is currently lacking.

## 2. Materials and Methods

### 2.1. Registration

This trial was prospectively registered under the title “Collagen Cross-Linking with Photoactivated Riboflavin (PACK-CXL) for the treatment of presumed infectious keratitis in dogs” at preclinicaltrials.eu. Last modifications to the protocol were made on 10 August 2022.

### 2.2. Hypothesis

PACK-CXL treatment will increase treatment success in cases of presumed infectious keratitis in dogs, compared to standard medical therapy.

### 2.3. Study Objectives

To assess treatment success proportions of PACK-CXL versus standard medical therapy in dogs with presumed infectious keratitis;To assess healing time, defined as the time elapsed between treatment start and complete epithelization of the ulcer;To assess costs (veterinary service and medication costs) of both types of treatment.

### 2.4. Trial Design

Prospective, multicenter, masked, randomized, controlled, interventional trial with two parallel and equally sized treatment groups.

### 2.5. Participants

#### 2.5.1. Eligibility Criteria

Trial enrolment is restricted to client-owned dogs that: Have a clinical diagnosis of a unilateral, presumed bacterial, infectious keratitis (loss of outer epithelium in combination with stromal loss and/or stromal neutrophil infil-trates; neutrophilic inflammation or bacteria confirmed on cytology);Have a maximal ulcer depth of 50% or less and a maximal ulcer diameter of 10 mm or less;Have an American Society of Anesthesiologists (ASA) grade I–III.Additionally, the owner needs to be able to comply with an intensive topical treatment schedule and give study consent (Appendix A).

There were no additional age, breed, or sex restrictions. Owners were given written information about the study and intervention in the form of a flyer (Appendix A).

Excluded from the study are dogs who: Suffer from a potentially immunosuppressive systemic disease (e.g., diabetes mellitus, Cushing);Take systemic or topical corticosteroid treatment;Suffer from concurrent ocular conditions that likely impair the healing potential of the cornea (e.g., dry eye, glaucoma).

#### 2.5.2. Study Setting

Patients are enrolled in eleven hospitals, including academic veterinary teaching hospitals and private veterinary practices (USA—1, France—2, Switzerland—1, Germany—1, Italy—1, Netherlands—1, United Kingdom—4). All participating clinicians are certified veterinary ophthalmologists or veterinary ophthalmologists in training. All collaborators received an online training explaining the study design and the use of the online data collection system in detail. Online meetings between collaborators will be organized every 3–4 months during the patient enrolment phase.

#### 2.5.3. Recruitment

Patients are recruited from cases referred to the collaborating ophthalmology centers. The trial was promoted in Switzerland, where information leaflets were sent to primary clinicians and veterinary ophthalmologists that were not directly involved in the trial, and in the United States, where the trial information was posted on the veterinary teaching hospital’s website and a mailing was sent out to primary clinicians and veterinary ophthalmologists that were not directly involved in the trial. In other collaborating centers, the trial was not promoted.

#### 2.5.4. Standardized Diagnostic Evaluation

Each patient will undergo a standard ophthalmic examination. Samples collected for this study are part of routine diagnostic testing for infectious keratitis. Samples are collected for: Bacterial culture and antibiotic sensitivity testing through application of a topical anesthetic to the affected eye (routinely used at the clinic), waiting for 30 s, and using a commercially available swab with transport medium to obtain a sample by gently rotating the tip of the swab on the edge of the ulcer;Cytology from the corneal ulcer area after obtaining a bacteriology sample, through gentle rotation of a cytobrush on the edge of the ulcer, performing a smear, and in-house staining with a diff quick stain. The presence and number of cells, cellular morphology, and presence and type of bacteria will be evaluated.

#### 2.5.5. Baseline Characteristics and Stratification

Information on baseline characteristics of dogs recruited to the study include patient age, gender, and breed. To maximize patient enrolment into the trial we will allow a next-day enrolment of patients that were seen as an out-of-hours emergency and that have received overnight treatment according to emergency clinic protocols. To avoid confounding, patients will be pre-stratified according to admission time:Routine admission: direct enrolment and randomization.Out-of-hours admission: next day enrolment and randomization.

### 2.6. Interventions

#### 2.6.1. Study Intervention and Standard Therapy

Dogs in the control arm will receive medical therapy consisting of systemic doxycycline, in combination with frequent topical antibiotic and anticollagenase eye drops. Animals in the intervention arm will receive PACK-CXL (sedation or general anesthesia are allowed if necessary) as antimicrobial and tissue-stabilizing treatment, permitting a reduction in antibiotic eye drop frequency to four times daily. Patients in both groups will receive systemic anti-inflammatory and pain medications and topical atropine as cycloplegic. Table 1 illustrates the treatment regimes for both groups.

#### 2.6.2. PACK-CXL (Intervention Arm) Procedure

Step 1—saturation of the corneal stroma with a commercially available riboflavin solution (Peschke M^®^ Riboflavin Solutions, Peschke GmbH, Huenenberg, Switzerland) through application of riboflavin eye drops every two minutes for 20 min; removal of excess riboflavin from the corneal surface prior to step 2. Measurement of corneal lesion size to adjust the illumination field diameter.

Step 2—illumination with UV-A: total energy delivery (fluence) of 16.2 J/cm^2^ with a fixed distance of 50 mm between energy source and corneal surface. Application of PACK-CXL in three cycles: 10 min irradiation at 9 mW, followed by two 2 min irradiations at 45 mW. Re-application of riboflavin drops between irradiation cycles for a one-minute corneal surface contact time, with removal of excess riboflavin prior to the next UV-A cycle.

#### 2.6.3. Criteria to Modify Allocated Treatment

Treatment may be modified in case of worsening of the clinical appearance:Increased ulcer surface area;Thinning of the residual stroma;Increased neutrophil infiltrate size and/or density;Liquefaction or corneal necrosis if not already present at initial examination.

If any of the above occurs, the clinician will be unmasked regarding patient allocation in order to make the best decision about necessary treatment modifications to stabilize the infectious keratitis. These patients will be classified as “treatment failure” during the primary data analysis. If culture/sensitivity testing and clinical examination results indicate that the initial antibiotic is ineffective, a change in antibiotic type will not be classified as treatment failure. Table 2 illustrates possible treatment modifications and consecutive actions and Figure 1 presents the study flow chart.

### 2.7. Measured Outcomes

Primary outcome: the number of cases successfully treated with the allocated treatment protocol.Secondary outcome is the time elapsed between treatment start and complete epithelization of the ulcer. We chose this outcome as re-epithelization is related to stromal stability and ulcer healing.Tertiary outcome: costs for veterinary services and medications.Fourth outcome: adverse reaction types and frequency in each treatment arm.

#### Participant Timeline

Ophthalmic re-examinations will be performed at days 7, 14, and 28 after trial enrolment (with a two-day buffer). Additional re-examinations are optional. As patients are often hospitalized during the initial days of treatment and assessment, data gathered during these hospitalization days can be collected on a voluntary basis. The maximum participation period in the clinical trial for an individual is 28 days, as illustrated in Figure 1. If a corneal ulcer has healed before day 28, or if the patient is classified as a treatment failure, this timepoint is considered as the end of the study for this individual patient. Patients that have not healed (continued stromal instability and/or presence of a fluorescein positive defect) at day 28 will be classified as “treatment failure” at study conclusion. The reason for treatment failure will be listed. These patients will be rechecked, treated, and followed outside the scope of the study until the infectious keratitis is resolved. Patients that are lost to follow up will be classified as “treatment failure”. Patients undergoing tectonic corneal surgery or enucleation, which precludes further monitoring of corneal healing, will be classified as “treatment failure” and their trial participation will stop. A schematic diagram according to the SPIRIT statement is presented in Table 3.

### 2.8. Sample Size

We calculated the sample size with the aim to detect a significant difference in the proportion of successes between the treatment and the control group using a one-sided hypothesis test and critical level of significance of 0.05. Based on a retrospective analysis of published and unpublished veterinary PACK-CXL cases [36], we assumed a 12% increase in treatment success, i.e., a 75% success in the control arm and an 87% success for the PACK-CXL intervention arm. The sample size calculation, involving a group sequential design, was performed following the approach of O’Brien and Fleming [37]. According to this method, the trial can be finished as soon as the null hypothesis is rejected in the first or second interim data analysis while controlling the alpha error at a nominal level of 5%. The two interim analyses will take place after the enrolment of 90 and 180 patients, respectively (Table 4). The sample size calculation and stopping rule were established with R software version 4.0.5, available online: https://www.r-project.org/ (accessed on 2 February 2019) [38], package “gsDesign”, available online: https://cran.r-project.org/web/packages/gsDesign/index.html (accessed on 2 February 2019) [39].

### 2.9. Randomization

#### 2.9.1. Sequence Generation

A random allocation sequence was generated with R package “blockrand”, available online: https://cran.r-project.org/web/packages/blockrand/index.html (accessed on 2 February 2019) [40], with a permuted block size of 2 to 4 patients, and uploaded to the electronic data capture system REDCap [41] by the trial coordinator (MK).

#### 2.9.2. Allocation

Patients will be randomly assigned per clinic to either the control or the intervention arm. Randomization and treatment allocation concealment will be performed via the electronic data capture system REDCap with trial management support from the Clinical Trials Center, University Hospital Zurich.

### 2.10. Masking

The presence of a shaved patch at the location of intravenous catheter placement typically is the only grossly visible difference between patients allocated to the intervention arm (PACK-CXL treatment performed in sedation or general anesthesia) or control arm. Therefore, all patients will receive a bandage covering a part of one of the front legs at the (potential) catheter placement site before entering the examination room. Patients will be re-examined by a clinician who was not involved in the initial patient enrolment and treatment procedure. In the majority of collaborating centers, patients are routinely examined by the ophthalmology trainee first and by the senior clinician afterwards. We will use this structure and ensure that at least one of the clinicians present at the re-examinations is masked to patient allocation. If the ophthalmology trainee is the masked clinician, the attending clinician will also fill out a trial patient chart with measurements and evaluations.

### 2.11. Data Collection, Management and Analysis

#### 2.11.1. Data Collection and Safety

Data will be collected via the electronic data capture system REDCap with trial management support from the Clinical Trials Center, University Hospital of Zurich (USZ). Data collection sheets are available at the Open Science Framework (Appendix A).

Appropriate coded identification (e.g., pseudonymization) is used in order to enter patient data into the database. All data entered are transferred to a mySQL database using encryption post filtering and sanitization to various relational database tables. The server hosting the REDCap database is maintained in an off-site locked server room. Only system administrators have direct access to the server and back-up tapes.

Access to the system and database is regulated by role-based user rights (e.g., investigator, statistician, administrator); this means that the access to various functions, such as the ability to export data, enter data, and see records, is controlled through the assigned user role. Entered data are only accessible and viewable by members from the institution at which the data was entered and not to members from other collaborating institutes.

A built-in data logging tool (audit trail) ensures that any changes to the project or user activity, including contextual information (e.g., the project record being accessed), are continuously tracked in real-time and accessible online or via a downloadable audit table.

A multi-level back-up system is in place. The whole system comprises internal back-ups, including the database back-up run several times per day. A daily external back-up is made in addition. The back-up tapes are stored in a secure place in a separate building.

The IT department of the USZ operates an Information Security Management System (ISMS) based on the ISO-27001 standard. This is an internationally recognized and established standard for information security.

When the trial is completed, the data set will be available at the Open Science Framework platform upon request.

#### 2.11.2. Data Analysis Plan

Potential confounding effects in this study will be controlled by restriction (see entry criteria), pre-stratification, and randomization. Results from both intention-to-treat (ITT) and per-protocol (PP) analyses will be reported.

Primary outcome analysis: the proportion of patients defined as treatment success in the control arm vs. interventional arm will be modeled with logistic regression where the stratification variable will be included. A mixed effects model will be considered to account for a potential clustering effect within clinical centers.

Secondary outcome data: time to re-epithelization. This variable is a repeat measurement at day 7, 14, and 28 with a binominal outcome (re-epithelialized: yes/no). To allow for the potential clustering within dog and clinic we will use a mixed model with patient ID/Clinic as a random effect.

Third outcome data: total treatment costs will be modeled with a linear regression or a mixed effects model with clinics as a random effect. Before being included in the model, the total cost per treatment will be adjusted for the purchasing power parity (PPP) and for the price estimate for a noncomplicated castration of a 5 kg dog given by each clinic. 

Further clinical examination data, including breed, age, cytology, corneal ulcer size, depth and stability data, and reported adverse reactions, will be evaluated using descriptive statistics and presented in tabular format.

Accumulated data will be monitored during planned interim analyses (after 90 and 180 enrolled patients) according to the group sequential design.

#### 2.11.3. Incomplete Data Sets

Patients with missing primary outcome-related data will be classified as “treatment failure”. The MissForest R package will be used to impute missing continuous measurements, under the assumption of missing at random. Categorical, missing data points that are not primary outcome-related will be removed from the tabular summary. The number of missing data points/incomplete data sets will be reported in the final manuscript.

### 2.12. Trial Monitoring

#### 2.12.1. Data and Safety Monitoring Board (DSMB)

A DSMB, consisting of two unbiased external consultants, will evaluate the results of the interim analyses at the two predefined interim analysis timepoints (after 90 and 180 enrolled patients). The DSMB will decide whether prematurely accumulated evidence for PACK-CXL superiority, or lack thereof, warrants premature trial termination. The DSMB will also be responsible for evaluating adverse effects documented during the trial. Based on that information and on the available literature, the DSMB will decide whether the trial needs to be terminated prematurely.

#### 2.12.2. Trial Stopping

The scheduled date for trial closure is December 2023. The randomized controlled trial can be stopped earlier than planned if the results of the planned interim analyses demonstrate a larger than expected benefit or harm of PACK-CXL. In addition, the randomized controlled trial may be prolonged if enrolment is insufficient for at least one interim analysis by December 2023.

## 3. Discussion

Striving for better reproducibility, the following are noteworthy recent innovations in scientific publishing: Requiring proof of adherence to study design and reporting guidelines;Pre-registration of study protocols. Both are mandatory in clinical trials reported in human medical journals, but not yet in preclinical research, including clinical studies involving animal patients. These innovations pave the way to transparent research, preventing a reporting and publication bias, and thus addressing the reproducibility crisis [42,43].

A randomized controlled trial is the most reliable method to determine whether an intervention is effective. Based on this study design, a cause-and-effect relationship between an intervention and disease outcome can be established [44]. According to a literature search (all publication abstracts and titles were screened), five randomized controlled trials [45,46,47,48] were published in the Journal of Veterinary Ophthalmology in 2021, out of 80 clinical studies (short reports and reviews were excluded). Information regarding study design, such as sample size calculation or masking, was missing in some publications.

Randomized controlled trials yield important information, but the findings can be misleading if study design and reporting are improperly performed. The inclusion of independent data and safety monitoring boards (DSMBs) is routine in pharmaceutical randomized controlled trials involving human patients, but not in veterinary medicine [49,50]. A DSMB typically includes clinicians and statisticians who are not directly involved in the trial. The major function of a DSMB is to oversee patient safety, primarily with respect to adverse events, and to review results from pre-planned interim analyses to determine whether a trial should be prematurely terminated.

Multicenter randomized controlled trials improve external validity through the enrolment of a study population that is more likely to be representative of the population at large [51]. Moreover, multicenter studies require rigorous study protocols to ensure uniform data collection across centers, which improves reproducibility. Unfortunately, multicenter randomized controlled trials involving veterinary patients are uncommon. According to Scimago, the *Journal of Feline Medicine and Surgery* (JFMS) and the *Journal of Small Animal Practice* (JSAP) are the small animal veterinary research journals with the highest impact. Based on the article abstracts from 2021, only 6.5% and 14.1% of all clinical research studies, published in JFMS and JSAP respectively, involved a multicenter collaboration, and none of these studies was a randomized controlled trial.

Clinical trial data are accumulated gradually, often over years. As a result, data from patients recruited early in the study are potentially available for interpretation while patient enrolment is still open. This provides a chance to use evidence emerging from the study to decide whether the study can and/or should be stopped early. For example, if a clear treatment difference is observed, stopping the trial is especially advantageous, since it prevents the assignment of additional patients to a therapy that has already been proven to be inferior. However, data analysis is traditionally performed after enrolment of the entire calculated sample size. The reason for conducting the analysis at the end relates to the frequentist approach. A predefined sample size allows the detection of clinical relevant difference between two treatment groups with a probability of 1 minus the s power. Unfortunately, when repeating significance tests, which happens when interim data analyses are performed, the type I error of the study is no longer controlled at the level of 5%.

The term “sequential design” refers to a group of statistical methods that involve repeated inspections of the trial data, replacing the fixed-sampled analysis, without compromising analysis validity [52]. Here, the significance level will be adjusted to account for the repetition (illustrated in Table 4). The essential idea behind it is to reduce the α level in the interim analyses so that the overall α remains below 5% [53]. Despite their great advantages, sequential methods are not often used in veterinary medicine. A reduction in patient numbers through early study conclusion without compromising validity (controlled family-wise error rate, no p-value stopping bias) is the main advantage of sequential approaches [52,53], also fostering the 3R principles. However, authors who choose to use a sequential method should be aware that the maximum sample size can be substantially larger than with a single stage design.

The estimation of the veterinary service and treatment costs, as assessed in this trial, does not necessarily reflect all costs carried by the owners. When complying with intensive medical treatment, owners may be forced to take vacation days or hire qualified pet care to apply the medications to their dogs at home, which generates an additional financial load. However, estimating veterinary service and medication costs is a parameter that may be useful when owners initially decide on the treatment option. Additionally, owners may feel compromised in other aspects of their life as a result of treatment, such as the time investment and stress on the animal–human bond involved, and their perception of how difficult/easy it is to comply with the veterinary treatment recommendations. These could also be considered as treatment costs and may be a factor in the decision-making process. Owner perceptions and needs receive little consideration in veterinary clinical studies, and deserve more attention.

In conclusion, a well-designed and executed randomized controlled trial will hopefully help answer the question of whether PACK-CXL treatment will increase treatment success in cases of stromal corneal ulcers in dogs, compared to standard medical therapy. The trial results will be reported as soon as the data analysis and manuscript have been finalized after successful trial conclusion. If PACK-CXL increases treatment success, PACK-CXL may become a routine clinical treatment modality with the potential to transform corneal ulcer treatment, decrease or replace antibiotic medical therapy, and increase treatment success rates and overall patient welfare, especially for brachycephalic patients. PACK-CXL thus has the potential to address important issues of the future, such as antibiotic resistance and access to specialty care, which will also contribute to human health.

## Figures and Tables

**Figure 1 animals-12-02862-f001:**
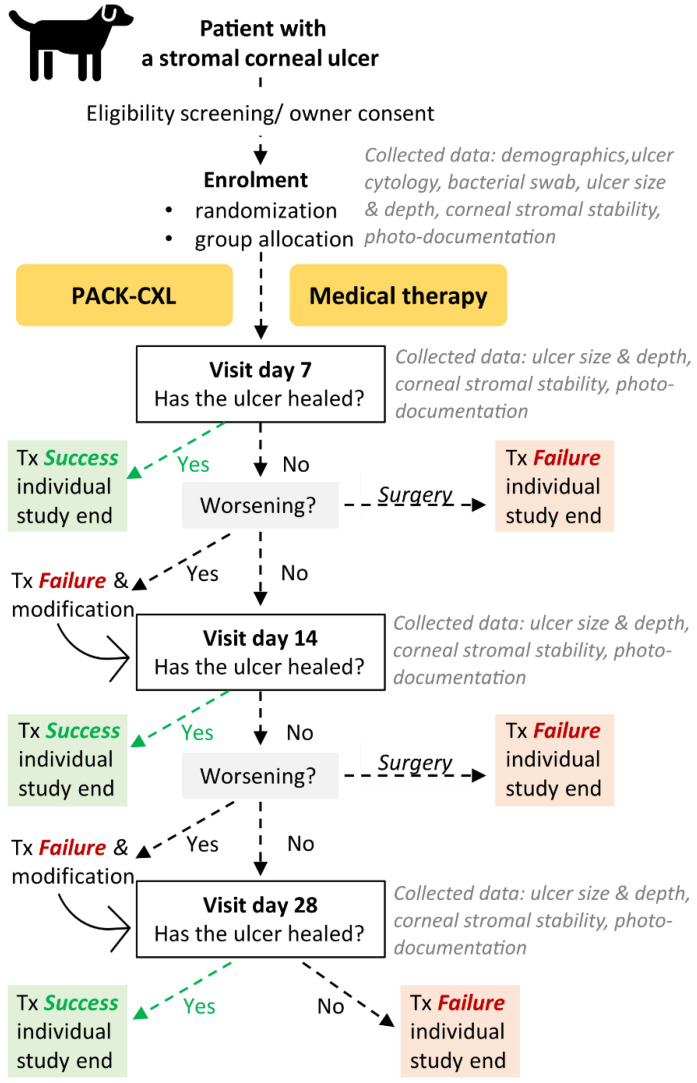
Study flow chart. Tx = treatment.

**Table 1 animals-12-02862-t001:** Medical treatment frequency and dosing.

Component	Active Ingredient	Intervention Arm (PACK-CXL)	Control Arm (Medical Management)
Topical antibiotics ^1^	Antibiotic treatment will be chosen based on geographic location, so that local bacterial sensitivities, drug availability and regulations can be met.	4 times daily	Every 2 h for 48 h (q 4 h at night), then reduce to 6 times daily
Anticollagenases ^2^	Serum/Plasma, EDTA, Acetylcysteine^2^	-	Every 2 h for 48 h (q 4 h at night), then reduce to 6 times daily
Systemic therapy	Doxycycline for 10 days	-	10 mg/kg SID per os
NSAIDs/ analgesia ^3^	Meloxicam 0.2 mg/kg on day 1 and 0.1 mg/kg afterwards	Once daily initially	
	Carprofen 2.2 mg/kg	Twice daily initially	
	Tramadol 2 mg/kg	Per os, up to 3 times daily,	
	Buprenorphine 0.02 mg/kg	Every 6–8 h as needed (hospitalized patients only)	
	Methadone 0.2 mg/kg	Every 4–6 h as needed (hospitalized patients only)	
	Paracetamol 10 mg/kg	Every 6–8 h as needed	
Cycloplegic ^3^	Atropine 1% Eye drops	As needed	
Tranquilizers ^3^	Will be chosen based on the clinician preferences and only if needed.		

^1^ Antibiotic therapy must be based on cytology findings. All animals should be treated with topical antibiotics until complete epithelial closure has occurred; ^2^ Use a minimum of one topical anticollagenase. All animals should be treated with topical collagenase-inhibitors until corneal stroma stability has been reached; ^3^ Choice of NSAID, opioid pain medication, cycloplegic, or tranquillizer at discretion of the clinician. A minimum interval of 10 min between application of different eye drops is required.

**Table 2 animals-12-02862-t002:** Possible modifications to the allocated treatment, consecutive actions, and patient classification in the records.

Allocated Treatment	Modification	Patient Classification	Consecutive Actions
Control arm (Medical management)	Increased antibiotic frequency	Failure	Data continue to be collected until ulcer re-epithelialization or to day 28
	Increased anticollagenase frequency	Failure	
	PACK-CXL	Failure	
	Change in antibiotic type based on laboratory results	Medical management	
	Ocular surgery	Failure	Study end for the individual
Intervention arm (PACK-CXL)	Increased antibiotic frequency	Failure	Data continue to be collected until ulcer re-epithelialization or to day 28
	Addition of anticollagenase treatment	Failure	
	Change in antibiotic type based on laboratory results	PACK-CXL	
	Ocular Surgery	Failure	Study end for the individual

**Table 3 animals-12-02862-t003:** Schematic timeline including the standard protocol items as defined by the 2013 SPIRIT statement.

		Study Period							
		Enrolment	Post-allocation						Closeout
		T0	T0	T1 *	T2 *	T3 *	T7	T14	T28
Enrolment	Eligibility screen ^1^	x							
	Ulcer cytology	x							
	Informed consent	x							
	Randomization	x							
	Treatment allocation	x							
	Photo-documentation ^2^	x							
Intervention	PACK-CXL procedure/ initiation medical management		x						
	Reduction in dosing frequency medical management				x				
Assessment	Bacteriology culture and sensitivity		x						
	Ophthalmic re-examination: ulcer size (mm^2^) ulcer depth (%) status of corneal stroma Photo-documentation ^2^			x	x	x	x	x	x

Eligibility screening ^1^ includes: (1) Medical interview to exclude dogs who suffer from a potentially immunosuppressive systemic disease, or take systemic or topical corticosteroid treatment; (2) General examination to assess ASA status. Only dogs with an ASA grade I–III will be included; (3) Ophthalmic examination based on which the dog is diagnosed with unilateral, presumed bacterial, infectious keratitis, and the status of the corneal stroma, ulcer size (mm^2^), and depth (%) are established and documented (only ulcers with a less than 10 mm diameter and less than 50% depth will be included). Dogs who suffer from concurrent ocular conditions that likely impair the healing potential of the cornea will be excluded. Photo-documentation ^2^: ulcer photography with a scale included in the picture (for example with Schirmer tear *test strip*); *: optional re-examinations.

**Table 4 animals-12-02862-t004:** Three stage group sequential design based on the method of O’Brien and Fleming [36].

Stops	Number of Recruited Patients	*p* Value
I	90	0.0015
II	180	0.0181
III	270	0.0437

## Data Availability

Not applicable.

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
