# Peer review of "Medical Management versus PACK-CXL in Dogs with Infectious Keratitis: A Randomized Controlled Trial Protocol"

_animals, 2022, doi:10.3390/ani12202862_

Round 1
Reviewer 1 Report
A well-designed and conducted randomized controlled trial is expected to answer the question of whether PACK-CXL treatment increases treatment success rates in cases of canine stromal corneal ulcers compared to standard medical therapy.The clinical trial protocol is the basis for planning, conducting and reporting the trial. It is important to resolve scientific, ethical, and safety issues prior to initiating a clinical trial. In addition, having the protocol published in this manner in advance may increase awareness of the trial and facilitate subject enrollment.
@ Recruitment and exclusion criteria @ Please provide clear recruitment and exclusion criteria into the study.
@ explanation and the consent form @ This clinical study is a study with intervention. Written explanations to owners are required. In addition, a consent form must be signed. Please present the written explanation of the clinical research to the owners and the consent form. You may register those documents as APPEDIX.
Author Response
Dear reviewer, we would like to thank you for your valuable comments and for taking the time to consider our manuscript. Below you can find information on how we addressed them.
@ Recruitment and exclusion criteria @ Please provide clear recruitment and exclusion criteria into the study
1) In line 148, (Participants/Eligibility criteria), we completed inclusion/exclusion criteria by adding: "There were no additional age, breed, or sex restrictions."
2) In line 168 (Recruitment), we completed information about the recruitment methods, which now reads: "Patients are recruited from cases referred to the collaborating ophthalmology centers. The trial was promoted in Switzerland, where information leaflets were sent to primary clinicians and veterinary ophthalmologists that were not directly involved in the trial, and in the United States, where the trial information was posted on the veterinary teaching hospital’s website, and a mailing was sent out to primary clinicians and veterinary ophthalmologists that were not directly involved in the trial. In other collaborating centers, the trial has not been promoted."
@ explanation and the consent form @ This clinical study is a study with intervention. Written explanations to owners are required. In addition, a consent form must be signed. Please present the written explanation of the clinical research to the owners and the consent form. You may register those documents as APPEDIX.
Those documents are available under Appendix A "Model consent form and information flyer given to the dog owners or authorized surrogates are available under DOI 10.17605/OSF.IO/YGWM8". Those documents are additionally referred to in the text in line 148: "The owner needs to be able to comply with an intensive topical treatment schedule and give study consent (Appendix A). There were no additional age, breed, or sex restrictions. Owners were given written information about the study and intervention in the form of a flyer (Appendix A)."
Reviewer 2 Report
The paper is very well written in English. The study protocol is detailed. The paper is a lit bit too long to highlight the importance of a clinical trial data analysis, but it is interesting.
However, for the lector the purpose is not clear in the beginning of the paper. I was expected to read about the results of the study too and not about only the study protocol of a multicenter study’s trial.
Page 1 line 23: Please add in the "Simple Summary" a line to clarify that the results of the trial will be expected only in a second time.
Author Response
The authors thank the reviewer for the favorable appraisal of the protocol.
Simple Summary lines 24, 25: We have added: "The trial results will be published after successful trial conclusion. "
Discussion lines 439-441: We have added: "The trial results will be reported as soon as the data analysis and manuscript have been finalized after successful trial conclusion."
Reviewer 3 Report
I commend the authors on their effort to establish and publish a solid clinical trial for dogs with infectious keratitis. The work is well written and the study design is well thought out, however I do have some comments for consideration:
Major comments
1) Anti-collagenolysis: Doxycycline is given to all patients and is considered by the authors under the category ‘anti-collagenolases agents’. There is no scientific evidence that this is true in dogs. In fact, the only two publications on this topic (Collins et al 2016; Sebbag et al 2018) showed that tear film concentrations are 1,000 to 10,000-fold below the levels required to inhibit corneal collagenolysis (≥ 455 μg/mL).
· Collins SP et al. Tear film concentrations of doxycycline following oral administration in ophthalmologically normal dogs. JAVMA 2016, 1;249(5):508-14
· Sebbag L et al. Impact of flow rate, collection devices, and extraction methods on tear concentrations following oral administration of doxycycline in dogs and cats. JOPT 2018;34(6):452-459.
If the authors would like to keep doxycycline in the study design, I suggest you re-brand it as ‘systemic antibiotherapy’ instead of anti-collagenolysis.
Similarly, the use of topical oxytetracycline cannot be included as ‘anti-collagenolysis’. The study by Sigmund et al on topical Terramycin® concluded that tear levels were higher than MIC for common bacterial pathogens, but also mentioned that “Anti-collagenolytic tear levels were not achieved at the timepoints evaluated or with the manufacturer-prescribed dosing frequency”.
2) Cost to owners: The authors fail to describe how cost to owners will be assessed. In my opinion, the direct ‘veterinary costs’ (initial visit, procedure, recheck visits, medications etc.) is not sufficient. When the required frequency of administration is q1-2h, many owners have to skip work or take vacation days to care for their dogs; this has a cost that could be grossly estimated by asking the owners in a questionnaire.
3) Assessed outcomes at each visit: I suggest including photo-documentation, as well as measuring the ulcer as width (mm) and height (mm) with overall surface described in mm2.
Other comments
· Line 12. Change ‘Corneal ulcers’ to ‘Infected corneal ulcers’. Otherwise the sentence is not scientifically correct, as a ‘corneal ulcer’ is not usually caused by bacteria in dogs, it is caused by other reasons and then gets complicated by bacterial invasion.
· Line 84. Add ‘are’ between pathogens and unable.
· Line 98: The authors mention ‘superiority’ to standard therapy. Isn’t it enough to show ‘equivalence’ or ‘superiority’ in order to promote PACK-CXL in veterinary medicine? In my opinion, an equivalent outcome with lower frequency of medications is still attractive to many dog owners.
· Lines 187-188: Revise this sentence as there is no evidence that systemic doxycycline helps as collagenolytic agent (See previous comment).
· Table 1. There is a typo: change ‘antybiotic’ to ‘antibiotic’. Also, place ‘Doxycycline’ in a separate category (something along ‘Systemic antibiotherapy’), and remove oxytetracycline.
· Figure 1. Typo ‘Patent’ should be ‘Patient’. Also consider adding photo-documentation, and reporting ulcers in mm2 and not mm.
Author Response
We would like to thank the reviewer for the thorough checking of our manuscript and for these very valuable comments.
Major comments
1) Anti-collagenolysis: Doxycycline is given to all patients and is considered by the authors under the category ‘anti-collagenolases agents’. There is no scientific evidence that this is true in dogs. In fact, the only two publications on this topic (Collins et al 2016; Sebbag et al 2018) showed that tear film concentrations are 1,000 to 10,000-fold below the levels required to inhibit corneal collagenolysis (≥ 455 μg/mL).
The decision about including Doxycycline was based on many discussions amongst all collaborators. Within the collaboration we reached a consensus decision to keep the tetracyclines in the medical treatment protocol (control group), since it is a part of the routine protocols in many of the collaborating clinics. However, we do agree with the reviewer regarding the lack of a scientific evidence base. We will table this topic for a renewed discussion with all the collaborators, also bringing up the references suggested by the reviewer. Unfortunately, we will not be able to organize a meeting with all collaborators within the period of time available for this review. We are very willing to change the initial treatment plan and acknowledge this deviation from the initial study protocol in the results manuscript. However, at the moment we prefer to include doxycycline as a part of the treatment to not change course more than once.
Similarly, the use of topical oxytetracycline cannot be included as ‘anti-collagenolysis’. The study by Sigmund et al on topical Terramycin® concluded that tear levels were higher than MIC for common bacterial pathogens, but also mentioned that “Anti-collagenolytic tear levels were not achieved at the timepoints evaluated or with the manufacturer-prescribed dosing frequency”.
Oxytetracycline was removed from the treatment protocol. (Table 1, line 206)
2) Cost to owners: The authors fail to describe how cost to owners will be assessed. In my opinion, the direct ‘veterinary costs’ (initial visit, procedure, recheck visits, medications etc.) is not sufficient. When the required frequency of administration is q1-2h, many owners have to skip work or take vacation days to care for their dogs; this has a cost that could be grossly estimated by asking the owners in a questionnaire.
We completely agree with the Reviewer. Owner perceptions and needs receive little consideration in veterinary clinical studies, and we would definitely like to see more research into these topics.
However, since the collaborating clinicians receive no financial or other support to participate in the trial, they have no incentive to engage in our trial, other than the rewards of contributing to evidence-based medicine. For this reason, and since trial patient enrolment is time-consuming and places much stress on already overbooked receiving schedules, we are concerned that additional data collection activities will potentially result in losing collaborator interest for the trial. Despite the fact that we think that this is a good idea, we would rather not add a client questionnaire to the trial protocol.
We have added this comment to the discussion section and changed the phrasing in the Introduction:
Introduction line 109-110: “Since pet owners are not required to purchase pet health insurance, the veterinary service and medication costs (initial visit, procedure, recheck visits, medications etc.) may be a crucial consideration.”
Materials and methods line 135: “To assess costs (veterinary service and medication costs) of both types of treatment.” Line 234, Measured Outcomes: “Tertiary outcome: costs for veterinary services and medications”
Discussion: line 423-434: “The estimation of the veterinary service and treatment costs, as assessed in this trial, does not necessarily reflect all costs carried by the owners. When complying with intensive medical treatment, owners may be forced to take vacation days or hire qualified pet care to apply the medications to their dogs at home, which generates an additional financial load. However, estimating veterinary service and medication costs is a parameter that may be useful when owners initially decide on the treatment option. Additionally, owners may feel compromised in other aspects of their life as a result of treatment, such as the time investment and stress on the animal-human bond involved, and their perception of how difficult/easy it is to comply with the veterinary treatment recommendations. These could also be considered as treatment costs and may be a factor in the decision-making process. Owner perceptions and needs receive little consideration in veterinary clinical studies, and deserve more attention.”
3) Assessed outcomes at each visit: I suggest including photo-documentation, as well as measuring the ulcer as width (mm) and height (mm) with overall surface described in mm2.
We incorporated these comments into the protocol. Removed SI units from Figure 1 and added SI units to Table 3 (line 235). Photo-documentation is now part of Figure1 and Table 3.
Other comments
- Line 12. Change ‘Corneal ulcers’ to ‘Infected corneal ulcers’. Otherwise the sentence is not scientifically correct, as a ‘corneal ulcer’ is not usually caused by bacteria in dogs, it is caused by other reasons and then gets complicated by bacterial invasion. Adopted
- Line 84. Add ‘are’ between pathogens and unable. Adopted
- Line 98: The authors mention ‘superiority’ to standard therapy. Isn’t it enough to show ‘equivalence’ or ‘superiority’ in order to promote PACK-CXL in veterinary medicine? In my opinion, an equivalent outcome with lower frequency of medications is still attractive to many dog owners.
We agree with the reviewer and believe that PACK-CXL would be an attractive treatment alternative for medical management even if the treatment outcomes are equal. However, even if PACK-CXL as a treatment alternative would be equivalent and not superior, we still prefer to use the superiority approach. Both study designs differ, and equivalence studies require larger sample sizes We are interested in detecting a relatively small difference in treatment success between PACK-CXL and medical management, namely 12%. We believe that the current design is a good compromise between realistic sample size, treatment effect, and obtaining evidence-based arguments for a wider acceptation of the use of PACK-CXL
- Lines 187-188: Revise this sentence as there is no evidence that systemic doxycycline helps as collagenolytic agent (See previous comment).
Lines 189-190 Adopted to ” Dogs in the control arm will receive medical therapy consisting of systemic doxycycline, in combination with frequent topical antibiotic and anticollagenase eyedrops”
- Table 1. There is a typo: change ‘antybiotic’ to ‘antibiotic’. Also, place ‘Doxycycline’ in a separate category (something along ‘Systemic antibiotherapy’), and remove oxytetracycline. Adopted. Doxycycline is listed as ‘Systemic therapy’
- Figure 1. Typo ‘Patent’ should be ‘Patient’. Also consider adding photo-documentation, and reporting ulcers in mm2 and not mm. Adopted
